# Interaction between Graphene Nanoribbon and an Array of QDs: Introducing Nano Grating

Sahar Armaghani [1], Ali Rostami [1] and Peyman Mirtaheri [2,*]

1   Photonics and Nanocrystal Research Lab. (PNRL), Faculty of Electrical and Computer Engineering, University of Tabriz, Tabriz 5166614761, Iran; s.armaghani@tabrizu.ac.ir (S.A.); rostami@tabrizu.ac.ir (A.R.)
2   Department of Mechanical, Electronics and Chemical Engineering, OsloMet—Oslo Metropolitan University, 0167 Oslo, Norway
*   Correspondence: peyman.mirtaheri@oslomet.no

**Abstract:** In this work, the interaction between an array of QDs and Graphene nanoribbon is modeled using dipole–dipole interaction. Then, based on the presented model, we study the linear optical properties of the considered system and find that by changing the size, number, and type of quantum dots as well as how they are arranged, the optical properties can be controlled and the controllable grating plasmonic waveguides can be implemented. Therefore, we introduce different structures, compare them together and find that each of them can be useful based on their application in optical integrated circuits. The quantum dot arrays are located on a graphene nanoribbon with dimensions of $775 \times 40$ nm$^2$. Applying electromagnetic waves with a wavelength of 1.55 µm causes polarization in the quantum dots and induces surface polarization on graphene. It is shown that, considering the large radius of the quantum dot, the induced polarization is increased, and ultimately the interaction with other quantum dots and graphene nanoribbon is stronger. Similarly, the distance between quantum dots and the number of QDs on Graphene nanoribbon are basic factors that affect the interaction between QDs and nanoribbon. Due to the polarization effect of these elements between each other, we see the creation of the effective grating refractive index in the plasmonic waveguide. This has many applications in quantum optical integrated circuits, nano-scale atomic lithography for nano-scale production, the adjustment coupling coefficient between waveguides, and the implementation of optical gates, reflectors, detectors, modulators, and others.

**Keywords:** total susceptibility; graphene nanoribbon; quantum dot; surface plasmon; bipolar interaction

## 1. Introduction

An effectively integrated waveguide is undoubtedly the basis for the development of photonic integrated circuits. Dielectric waveguides operate on the principle of total internal reflectance (TIR) to guide the propagating light in a waveguide core with a high refractive index over long distances with low losses of ~3–10 dB/m [1]. However, in these cases, the diffraction of light as a limiting point should be considered. In contrast, plasmonic waveguides emit a wave of surface plasmons or surface plasmon polaritons (SPPs) at the dielectric-metal interface which is increasingly limited in the direction perpendicular to the propagation [2,3]. Recent reports of plasmonic waveguides, such as metal-dielectric-metal (MDM) [4,5], dielectric-metal-dielectric (DMD) [6], hybrid surface plasmon polariton (HSPP) [7], and dielectric loading surface plasmon polariton (DLSPP) [8] has attracted considerable attention. Plasmonic waveguides can enhance light-matter interactions [9]. However, due to the existence of free electrons at the boundary between metal and the dielectric waveguides with ohmic losses capability, the diffusion losses will be considerable. Achieving the dense integration of the waveguide systems is impossible considering the diffraction problem [4]. All-graphene-based technologies are suitable approaches to solve this problem because the electromagnetic field localizes well to graphene [10,11]. Recently,

graphene, a one-atom-thick material, has been introduced as a plasmonic planar material [12,13]. Thus, graphene plasmons are an attractive and suitable alternative to noble metal plasmons because they show a relatively large distance for plasmon transmission. In addition, surface plasmons in graphene have the advantage of being regulated by electrostatic gates [10,14]. Compared to noble metals, graphene has extraordinary electronic and mechanical properties that originate in part from its zero-mass charge carriers [15]. Hybrid plasmonic structures, while combining the properties of dielectric and plasmonic waveguides, are designed to achieve high light confinement without including high losses [16]. Experimental research has expanded to include the fabrication and study of QD-graphene nanostructures [10,17]. These structures can make a significant contribution to the formation of plasmonic waveguides. In integrated photonic circuits based on surface plasmon polariton, graphene as a high-speed substrate to transfer the carriers and quantum dots as an information processing unit can be merged to become an excellent candidate to replace integrated electronic circuits [18,19]. Applying electromagnetic waves to the surface of graphene nanorods creates the surface plasmon. The plasmon induced in the graphene nanoribbon excites the exciton at the quantum point and changes the electron distribution in the QDs, thus inserting a dipole at a quantum point. Thus, it creates a bipolar electric field on the nano-graphene strip and the quantum dot. The electric field created by the quantum dot on the graphene nanorod plate is due to the QD polarization caused by surface plasmons on the graphene. Numerical simulations show that the periodic optical structure is obtained by placing quantum dots on a graphene nanorod.

This paper calculates the optical properties, including the absorption coefficient and refractive index of a graphene nanoribbon and an array of QDs on it. This analysis studies the effect of the geometrical and optical parameters of the QDs and the effect of graphene nanoribbon on their electrical and optical properties. The mathematical modeling, including analytical and numerical approaches, is presented and discussed in Section 2. Finally, the effect of different arrays of the QDs on graphene nanoribbon on optical properties is investigated and discussed in Section 3.

## 2. Mathematical Formalism

### 2.1. Induced Polarization in Graphene and Quantum Dots

By considering the permittivity of metals based on the plasma model, it is clear that there are different behaviors in different radiant frequencies (from pure metal to dielectric). When the Fermi energy level of graphene leaves the Dirac point, then, due to the lack of a bandgap between the conduction and valance bands, the behavior of graphene is the same as pure metals (gold and silver, etc.). However, graphene's highly dispersive behavior should be considered. The above-mentioned points suggest that since the relaxation time of the electron in this material is in the picosecond range, the far-infrared spectrum for graphene is a booklet of low frequencies. Therefore, this material operates as a dielectric with positive permittivity at any Fermi energy level [20]. Based on the physical properties of plasmonic materials, the number of dipoles formed at the surface is given. A harmonic oscillator differential equation is used to compute the polarization caused by an external electric field applied in a plasma model. By obtaining the amount of electron displacement from the steady-state solution of the harmonic oscillator differential equation, we can calculate the polarization of the graphene nanoribbon. Thus, the polarization of graphene is given as follows [21]

$$\widetilde{p}^g = -\frac{\varepsilon_0 \omega_p^2(\mu_c)}{(\omega^2 + i\gamma\omega)} \vec{E} = -\xi(\omega, \mu_c) \vec{E} \tag{1}$$

where $\gamma$ and $\xi(\omega, \mu_c) = \varepsilon_0 \omega_p^2(\mu_c)/(\omega^2 + i\gamma\omega)$ are the inverse of the electron relaxation time and a new function appears in Equation (1), respectively.

The Plasma frequency ($\omega_p$) is the frequency when the real part of the permittivity of matter becomes zero. The plasmonic behavior of metals results from a change in their

dielectric coefficient concerning the radiant frequency. Therefore, the Plasma frequency of graphene is also a significant parameter in the design of various devices. On the other hand, graphene has the property of adjusting the Fermi surface so it is possible to change the plasma frequency by changing the chemical potential ($\mu_c$) of this material. This feature in graphene increases the importance of this material in the design of plasmonic devices. The free-electron relaxation time ($\tau = \gamma^{-1}$) is twice the scattering rate of the charged particle (2$\Gamma$). This is proportional to graphene's Fermi velocity ($v_f = 10^6$ m·s$^{-1}$), chemical potential, and electron excitability ($\mu_n \approx 4$ m$^2$·V$^{-1}$·s$^{-1}$). Electromagnetic waves in a graphene nanoribbon cause electrons to move in the opposite direction of the electric field, forming a dipole moment with the center of the atoms being positive and constant. Therefore, the descending electric field ($E_x = E_0 \cos(\omega t)$) also makes up surface polarizations on the graphene nanoribbon. A bipolar electric structure introduces an electric field in its surroundings on graphene nanoribbon [21,22] and is calculated separately from the main field ($E^{dipole} = \frac{1}{4\pi\varepsilon_b R^3}\left(3\frac{\vec{R}.\vec{P}}{R^2}\vec{R} - \vec{P}\right)$, where $\varepsilon_b$, $R$ and $P$ are the permittivity of the medium surrounding the QD-graphene system, the distance between the center of polarization and a desire for point, and the polarization of the quantum dot and graphene nanoribbon, respectively). Therefore, the dipole field created by graphene nanoribbons at the center of the quantum dot can be calculated in three directions, obtained from surface plasmons. Therefore, the dipole field created by its orbiting graphene nanoribbons can be calculated in three directions as below. These are obtained from surface plasmons, if the length of graphene nanoribbons is in the *X*-axis on the *X-Y* plane (see Figure 1).

$$E_{\text{int}}^{g-} = \frac{\xi(\omega)E_0}{4\pi\varepsilon_b} \times \frac{1}{R_Q^3}\left(3\left(\frac{x_Q}{R_Q}\right)^2 - 1\right)a_x + \frac{3\xi(\omega)E_0}{4\pi\varepsilon_b R_Q^5}z_Q x_Q a_z \tag{2}$$

where $R_Q$ is the distance between the desired location on graphene nanoribbon and the center of the quantum dot. Moreover, $a_x$, $a_y$, and $a_z$ are basic unit vectors in the coordinate.

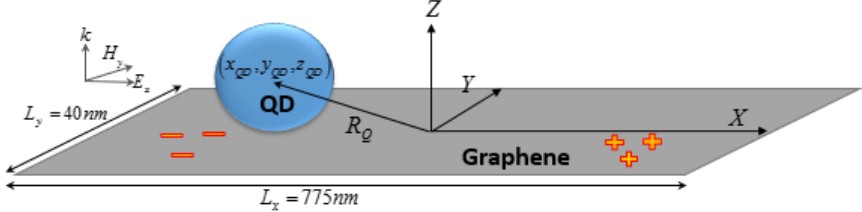

**Figure 1.** The structure under the study with induced polarization [23].

This field is accounted for while the number of polaritons formed on the surface of the graphene nanoribbon is one. Where $R_Q$ represents the distance between the axis of polarization of the surface and the desired point ($x_Q, y_Q, z_Q$) (see Figure 1). Similarly, the dipole created in the quantum dot induces an electric field around itself. Considering Columbus's law and incident applied to the electric field, the induced electric field in the surrounding medium by polarization in QD ($\vec{P}^{QD} = \widetilde{p}_x^{QD}\hat{X}$) is obtained as follows.

$$E_{\text{int}}^{QD-} = \frac{3\widetilde{p}^{QD}}{4\pi\varepsilon_b R^3}\left(\left(\frac{x_Q - x_{QD}}{R}\right)^2 - \frac{1}{3}\right)a_x + \frac{3\widetilde{p}^{QD}}{4\pi\varepsilon_b R^5}(x_Q - x_{QD})(y_Q - y_{QD})a_y + \frac{3\widetilde{p}^{QD}}{4\pi\varepsilon_b R^5}(x_Q - x_{QD})(z_Q - r_{QD})a_z \tag{3}$$

If there is a neighboring array of quantum dots, the field affected by them at a point ($x_Q, y_Q, z_Q$) is equal to the sum of the field produced by each of them.

### 2.2. Interaction between Graphene Nanoribbon and Quantum Dots

In this paper, we introduce different structures of an array of quantum dots on the graphene nanoribbon and model the interaction between them. The electric field acting

on the graphene plate is the sum of the applied field ($E_x$) and the field results from the polarization of quantum dots ($E_{\text{int}}^{QD-g}$). The polarization in quantum dots originates from a set of three fields.

The first component is the applied electric field ($E_x$), the second component is the induced electric field due to the effect of the polariton field of the graphene surface Plasmon ($E_{\text{int}}^{g-QDj}$), and the third component is the induced field due to the polarization field of interaction between quantum dots ($E_{\text{int}}^{QD-QDj}$). Thus, the total field is the sum of the three components and is given as follows.

$$E^g = \left( E_x + E_{\text{int}}^{QD-g} \right)$$
$$E^{QD_j} = \frac{\left( E_x + E_{\text{int}}^{g-QD_j} + E_{\text{int}}^{QD-QD_j} \right)}{\varepsilon_{bd}} \tag{4}$$

Here, $\varepsilon_{bd} = (2\varepsilon_b + \varepsilon_d)/3\varepsilon_b$ and $\varepsilon_d$ is the permittivity of the quantum dot [20]. To study the interaction between particles, at first, the interaction for a single quantum dot is modeled and then the mutual effect of the quantum dots is considered. Therefore, according to the boundary conditions ($x_g = x_Q, y_g = y_Q$), the tangential electric fields between these two materials must be equal, and the amount of initial polarization induced at the quantum dots is calculated as follows.

$$E_t^{QD} = E_t^g \rightarrow E_x^{QD} = E_x^g \Rightarrow \frac{1}{\varepsilon_{bd}} \left( 1 + \frac{\zeta(\omega,\mu_c)}{4\pi\varepsilon_b} B^{QD-g} \right) E_0 = E_0 - \frac{\widetilde{p}^{QD}}{4\pi\varepsilon_b r_{QD}^3}$$
$$\widetilde{p}_i^{QD} = 4\pi\varepsilon_b r_{QD}^3 \left( \varepsilon_m - \frac{\zeta(\omega,\mu_c)}{4\pi\varepsilon_b\varepsilon_{bd}} B^{QD-g} \right) E_0, \varepsilon_m = 1 - \frac{1}{\varepsilon_{bd}} \tag{5}$$

The constant $B^{QD\text{-}g}$ shows the effect of the polarization of surface plasmons on the induced polarization of quantum dots. This is called the 'strength of interaction' between the quantum dots and the graphene nanoribbon and is given as ($B^{QD-g} = (-1/R_{c-c}^3)\left(3(x_Q/R_{c-c})^2 - 1\right)$), where $R_{c-c}$ represents the distance from the graphene nanoribbon center to the quantum dot center. As an exceptional case, we consider a single QD on a graphene nanoribbon, and the total polarization, including the polarization due to be applied to the electric field and induced by graphene nanoribbon in a different position is calculated. It is shown that the displacement of QD on graphene nanoribbon does not considerably change the polarization.

As demonstrated, the displacement of the quantum dot does not have a significant effect on the polarization of the quantum dot, but the change in radius has a relatively strong effect on the quantity of the quantum dot polarization (Figure 2). Therefore, Figure 3 shows the polarization of the quantum dot affected by a graphene surface plasmon polariton based on the change in the radius of the quantum dot.

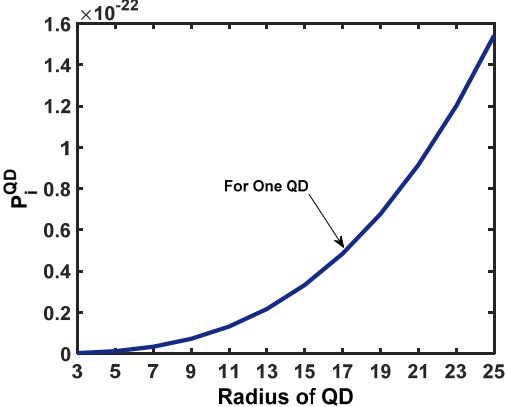

**Figure 2.** The effect of the radius of the quantum dot on the polarization (input light wavelength, the permittivity of the quantum dot, and chemical potential are $\lambda_{eff} = 1.55$ μm, $\varepsilon_d = 12$, $\mu_c = 0.3$ eV, respectively).

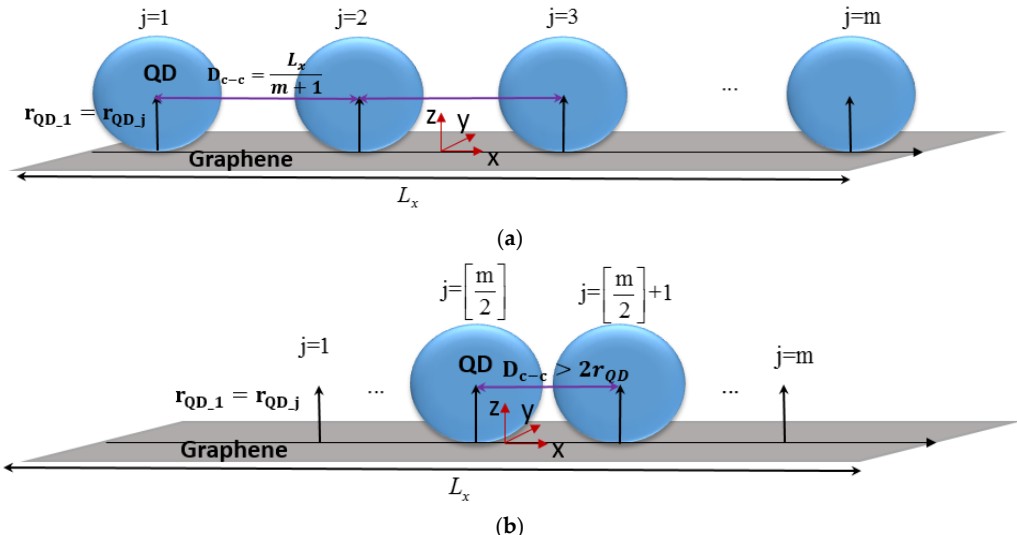

**Figure 3.** Quantum dots distribution with equal radius. In the first case (**a**) QDs distribute on the whole length of a graphene nanoribbon with the same distance. In the second case (**b**) QDs distribute on the length of a graphene nanoribbon but with a smaller same distance around the center of the graphene.

### 2.3. Interaction between Quantum Dots in an Array

It is important to consider how the quantum dots interact with each other to determine how they should be arranged to have effective interaction with the graphene nanoribbon. In this work, we use two models of quantum dots for an array. In the first model, all quantum dots have the same radius, and in the second model, it is assumed that all QDs have a different radius. As demonstrated in Section 2.2, if quantum dots have various radii, their polarization will be different. Otherwise, they will have the same polarization. Thus, we must consider these two models separately.

2.3.1. Array of Quantum Dots with the Same Radii

We also examine the first model in two cases. In the first case, we distribute the quantum dots considering their number along the central length of the nanoribbon with the same distances as shown in Figure 3a. In the second case, we distribute the quantum dots at a certain distance (smaller than the first case) from each other from the center to the edges of the graphene nanoribbon without considering their number. As it is shown in Figure 3b, the density of quantum dots is higher at the center of the longitudinal axis of the graphene nanoribbon.

Equation (3) represents the field that results from the polarization of a quantum dot. Due to this equation, the difference between the two cases of the first model is the distance that quantum dots have from each other. By superimposing the fields resulting from the polarization of other quantum dots on the center of a quantum dot, we can obtain the interaction of the quantum dot with other quantum dots. For the first case, the interaction of the array of quantum dots can be calculated by the following approach.

If quantum dots are distributed with a certain distance (D) on the longitudinal axis of graphene nanoribbon, their location is equal to $x_{QD1}, x_{QD1} + D, ..., x_{QD1} + (m-1)D$. By placing this relation in Equation (3), the following equation is obtained.

$$E_{\text{int}}^{QD_n - QD_j} = \left| \frac{\widetilde{p}^{QD}}{4\pi\varepsilon_b D^3} \right| B^{QD_n - QD_j} a_x \tag{6}$$

Now we need to introduce another interaction constant $B^{QD_n - QD_j} = 2 \times \sum\limits_{j=1, \neq n}^{m} (1/|j-n|^3)$.

The constant $B^{QD_n - QD_j}$ is equivalent to the interaction between quantum dots and shows

the effect of other QDs on the polarization of quantum dot number n. According to Equations (3) and (4), the electric field on the graphene nanoribbon $(x_g, y_g, 0)$ is given as follows.

$$E^g = \begin{pmatrix} a_x & a_y & a_z \end{pmatrix} \begin{pmatrix} \left| \frac{3\widetilde{p}^{QD}}{4\pi\varepsilon_b} \right| \sum_{j=1}^{m} \frac{1}{R_{g-c_j}^3} \left( \left( \frac{x_{QD1}+(j-1)D-x_g}{R_{g-c_j}} \right)^2 - \frac{1}{3} \right) + E_0 \\ \left| \frac{3\widetilde{p}^{QD}}{4\pi\varepsilon_b} \right| y_g \sum_{j=1}^{m} \frac{1}{R_{g-c_j}^5} \left( x_{QD1}+(j-1)D-x_g \right) \\ \left| \frac{3\widetilde{p}^{QD}}{4\pi\varepsilon_b} \right| r_{QD} \sum_{j=1}^{m} \frac{1}{R_{g-c_j}^5} \left( x_{QD1}+(j-1)D-x_g \right) \end{pmatrix} \quad (7)$$

where $R_{g-c_j}$ is the distance between the target position on the graphene nanoribbon and the center of quantum dot number j and is $R_{g-c_j} = \sqrt{(x_g - (x_{QD1}+(j-1)D))^2 + r_{QD}^2 + y_g^2}$. In addition, according to Equations (2), (4) and (6), the total electric field on the center of the quantum dot is given as follows.

$$E^{QD_n} = \begin{pmatrix} a_x & a_y & a_z \end{pmatrix} \begin{pmatrix} \frac{1}{\varepsilon_{bd}} \left( 1 + \frac{\zeta(\omega,\mu_c)}{4\pi\varepsilon_b} B^{QD_n-g} \right) E_0 + \left| \frac{\widetilde{p}^{QD}}{4\pi\varepsilon_b D^3} \right| B^{QD_n-QDj} \\ \frac{3\zeta(\omega,\mu_c)E_0}{4\pi\varepsilon_b R_{c-c_n}^5} y_{QD_n} x_{QD_n} \\ \frac{3\zeta(\omega,\mu_c)E_0}{4\pi\varepsilon_b R_{c-c_n}^5} r_{QD} x_{QD_n} \end{pmatrix} \quad (8)$$

Boundary conditions must be met for each quantum and graphene nanoribbon. Based on this, we can obtain the polarization of each quantum dot considering Equation (5).

$$\widetilde{p}_T^{QD} = \frac{4\pi\varepsilon_b \left( \frac{\zeta(\omega,\mu_c)}{4\pi\varepsilon_b\varepsilon_{bd}} B^{QD_m-g} - \varepsilon_m \right)}{3\sum_{j=1}^{m} \left( \frac{1}{R_{g-c_j}^3} \left( \left( \frac{(j-1)D}{R_{g-c_j}} \right)^2 - \frac{1}{3} \right) \right) - \left( \frac{B^{QDm-QDj}}{D^3} \right)} E_0 \quad (9)$$

As shown in Equation (9), the polarization of quantum dots is equal and depends on the distance between the quantum dots and their number. Figure 4 has colors in both red and blue. The red diagram shows the first case of the first model, and the number of quantum dots increases. The blue diagrams show the second case of the first model, where the distance between the quantum dots starts at $2.5r_{QD}$ and increases with the multiple of $\frac{r_{QD}}{2}$.

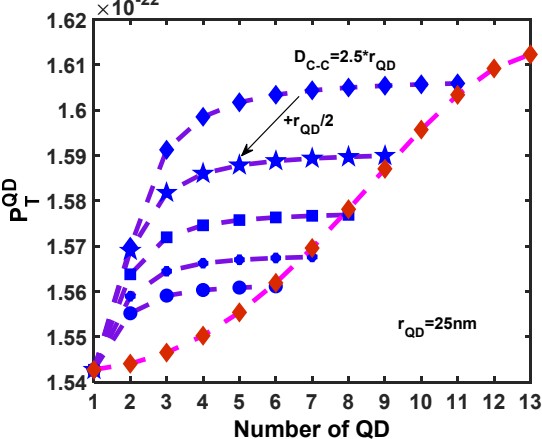

**Figure 4.** The ratio of polarization of quantum dots to their number in two states. Blue is for QDs with the same distance but around the center of the graphene nanoribbon and red is for QDs with the same distance but distributed in the whole length of the graphene nanoribbon ($L_g = 775$ nm, $W_g = 40$ nm, $\mu_c = 0.3$ eV, $\lambda_{eff} = 1.55$ μm).

### 2.3.2. Array of Quantum Dots with the Same Radii

As we showed in the previous sections, varying the radius of the quantum dot has a significant effect on their polarization. The second model of the array of quantum dots on the graphene nanoribbon is discussed in this section. The second model also has two cases, including the Gaussian and the sagittal distribution. Figure 5 shows the structure of these two modes.

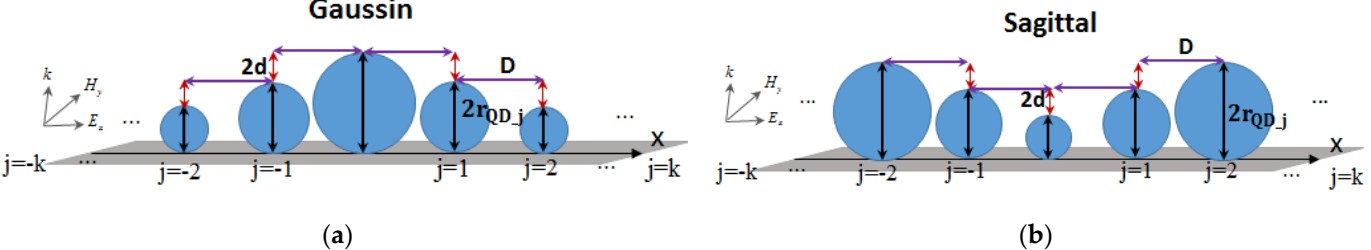

**Figure 5.** Quantum dots distribution, (**a**) the Gaussian distribution, and (**b**) the sagittal distribution.

Since the radius of the quantum dots is varied in the array, their initial polarization will be different. We calculated the initial polarization from the total field, including graphene's surface polarization and the incident wave (described in Section 2.2). If we distribute the $m = 2k + 1$ number of quantum dots with a certain distance (D) on the longitudinal axis of graphene nanoribbons as the Gaussian or sagittal type distributions, the location of QDs is $x_{QDm}, x_{QDm} \pm D, ..., x_{QDm} \pm (k-1)D, x_{QDm} = 0$. In addition, their radius for the Gaussian case is equal to $r_{QD_m}, r_{QD_m} - d, ..., r_{QD_m} - (k-1)d$, and for the sagittal case, the radius is $r_{QD_m}, r_{QD_m} + d, ..., r_{QD_m} + (k-1)d$. By putting their location in Equation (3), the array of quantum dots-interaction constant that appeared in Equation (8) is revised as follows.

$$B^{QDn-QD_j} = 2 \times \sum_{\substack{j=-k \\ j \neq n}}^{k} \frac{\widetilde{p}_{i-j}^{QD}}{\left(((j-n)D)^2 + ((|n|-|j|)d)^2\right)^{3/2}} \tag{10}$$

$\widetilde{P}_{i-j}^{QD}$ represents the initial polarization of the $j$th quantum dot. Each quantum dot has a permittivity that creates its overall polarization due to the effective fields at the center of the quantum dot. The total polarization of each quantum dot is given as follows.

$$\widetilde{P}_T^{QDn} = \varepsilon_0(\varepsilon_d - 1)E^{QDn} = \varepsilon_0(\varepsilon_d - 1)\left(\frac{1}{\varepsilon_{bd}}\left(1 + \frac{\xi(\omega, \mu_c)}{4\pi\varepsilon_b}B^{QDn-g}\right)E_0 + \left|\frac{\widetilde{p}^{QD}}{4\pi\varepsilon_b D^3}\right|B^{QDn-QD_j}\right) \tag{11}$$

According to Equations (3) and (4), the total fields on any point of the graphene nanoribbon $(x_g, y_g, 0)$ is given as follows.

$$E^g = \begin{pmatrix} a_x & a_y & a_z \end{pmatrix} \begin{pmatrix} \frac{3}{4\pi\varepsilon_b} \sum_{j=1}^{m} \frac{\widetilde{p}_T^{QDn}}{R_{g-c_j}^3}\left(\left(\frac{(j-1)D-x_g}{R_{g-c_j}}\right)^2 - \frac{1}{3}\right) + E_0 \\ \frac{3y_g}{4\pi\varepsilon_b} \sum_{j=1}^{m} \frac{\widetilde{p}_T^{QDn}}{R_{g-c_j}^5}((j-1)D - x_g) \\ \frac{3r_{QD}}{4\pi\varepsilon_b} \sum_{j=1}^{m} \frac{\widetilde{p}_T^{QDn}}{R_{g-c_j}^5}((j-1)D - x_g) \end{pmatrix} \tag{12}$$

According to Equation (4), concerning the degree of polarization obtained at the quantum dots, the total field induced on the graphene nanoribbon is also obtained. While the total susceptibility of the system is obtained using graphene polarization. The following equation expresses the total susceptibility of the structure.

$$\widetilde{p}^g = \varepsilon_0 \chi_e E^g \rightarrow \chi_e = \left( \frac{-\xi(\omega)}{\varepsilon_0} \right) \left( \frac{E_0}{E^g} \right) \tag{13}$$

As it is well known, the susceptibility shows the interaction of the electromagnetic field with the matter and so it is useful in optical waveguide design.

## 3. Results and Discussion

In this work, we used a graphene nanoribbon with dimensions of $775 \times 40$ nm$^2$ and an array of quantum dots with a permittivity of 12, and different radii and arrangements on it. We investigated the optical properties of the proposed structures for different arrangements. To this end, we considered two cases. At first, a periodic array with the same QDs' radius was considered. In the second case, we considered the different radii of the quantum dots. Incident electromagnetic wave at 1.55 μm irradiated on the structure. First, the results of the first case are considered and discussed. In this case, several arrangements of QDs can be considered. In general, we can study the first case by exploring two possibilities. In the first part, some QDs distribute uniformly in the length of the graphene nanoribbon. In the second case, QDs with a given distance between them are distributed around the nanoribbon center along the graphene length. Figure 5 compares the two cases.

According to Figure 6, we find that the presence of quantum dots makes susceptibility to change. Such a feature allows us to place an array of quantum dots on a graphene nanoribbon, and an optical grating is implemented. The grating blue and red periodic parts depend on the radius and distance between each quantum dot. It should be noted that the periodic arrangement of the quantum dots on the longitudinal axis of the graphene nanoribbon is desired. The red graphs show the concentration model of the quantum dots at the center of the graphene nanoribbon. Besides, the blue graph represents the uniform rotation of the quantum dots across the entire length of the graphene nanoribbon. In Figure 7, the dependency of the radius of the quantum dots to the grating width is illustrated.

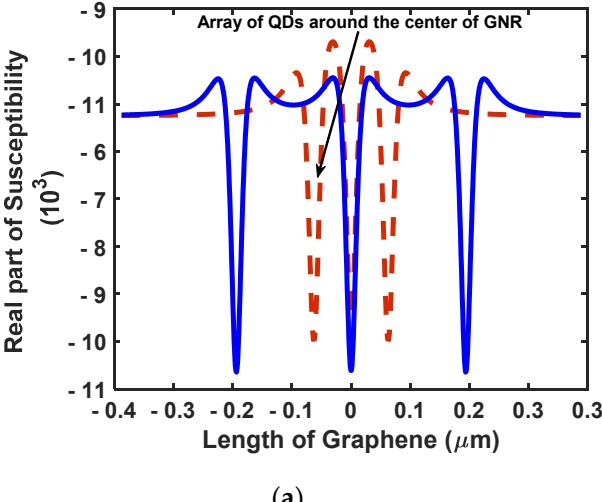

(**a**)

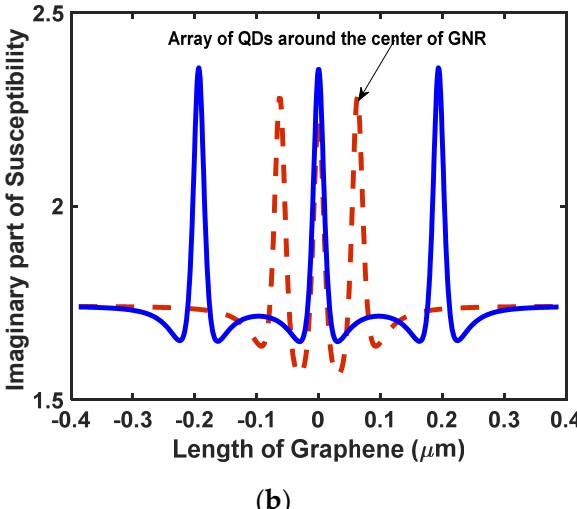

(**b**)

**Figure 6.** Susceptibility for the proposed system in (**a**) the real part of the susceptibility for an array of QDs distributed on the graphene nanoribbon with the same distance in the whole ribbon with a chemical potential of 0.9 eV $m = 3$, $r_{QD} = 25$ nm, and (**b**) the imaginary part of the susceptibility for an array of QDs distributed on the graphene nanoribbon with the same distance but centered around the center of the ribbon with $D_{C-C} = 3r_{QD}$, $r_{QD} = 25$ nm. The distance between QDs in red and blue curves are 45 nm and 194 nm, respectively.

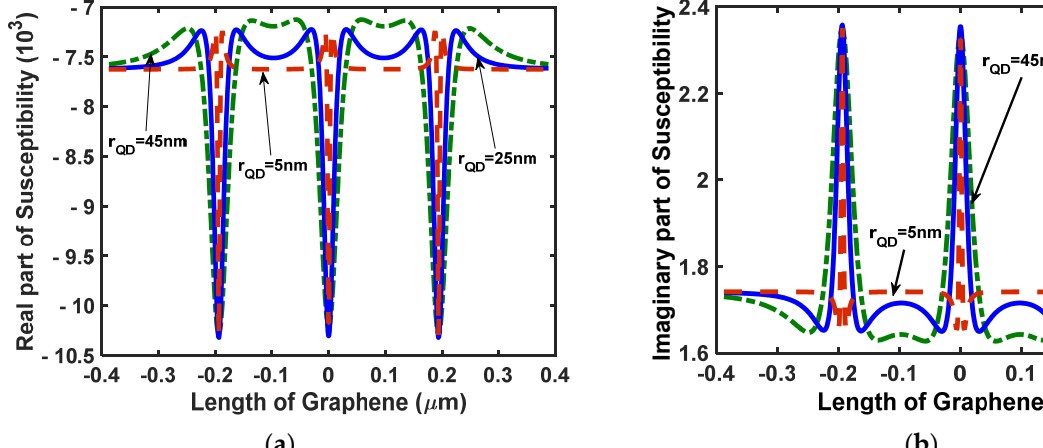

(**a**)                                           (**b**)

**Figure 7.** In the first case of the first model, the effect of quantum radius on (**a**) real part and (**b**) imaginary part of the susceptibility, $\mu_c = 0.9$ eV.

As shown in Figure 7, the grating width is broadened as the quantum dot radius increases. If the radius of the quantum dots is randomly chosen, the susceptibility of the proposed system will be interesting for many applications. Therefore, we introduced the second model with two Gaussian and sagittal cases. According to this point, Table 1 shows the radii used in these structures with their initial polarization.

**Table 1.** Initial polarization in terms of the radius of QD (nm).

| The Radius of QD (nm) | Initial Polarization $(\widetilde{P}_i^{QD}) \times 10^{21}$ |
|---|---|
| 10 | 0.0099 |
| 15 | 0.033 |
| 20 | 0.079 |
| 25 | 0.1543 |

Figures 8 and 9 show the susceptibility of the Gaussian and sagittal cases, respectively.

As it is clear, decreasing or increasing the radius of the quantum dots has a significant effect on the nano-grating implemented by the proposed structure. It makes the manufacturing industry of such waveguides more accurate. This is because most changes occur at the point where the largest quantum dot is located (See Figure 9).

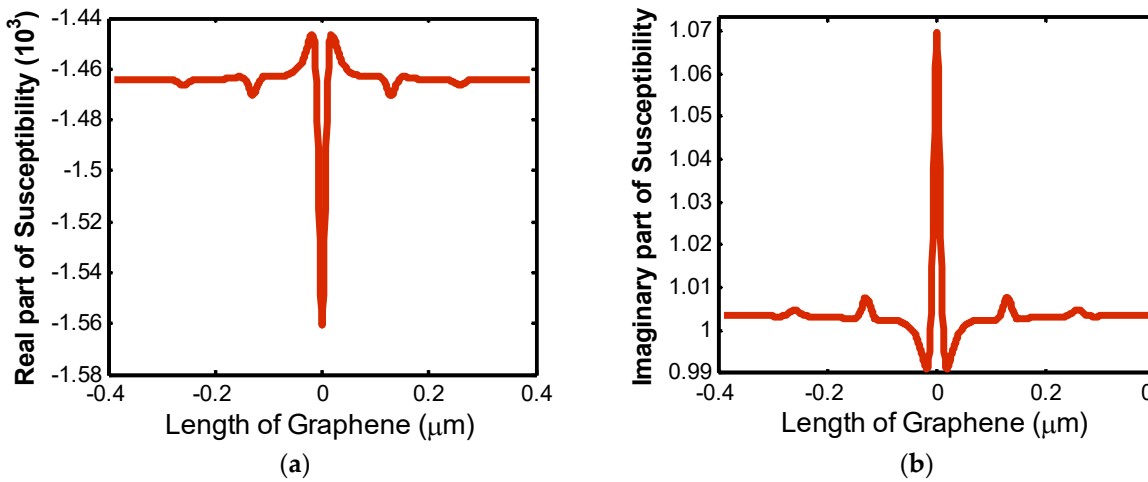

(**a**)                                           (**b**)

**Figure 8.** The susceptibility for the Gaussian case for distribution in whole graphene nanoribbon (**a**) real part, and (**b**) imaginary part, $\mu_c = 0.3$ eV.

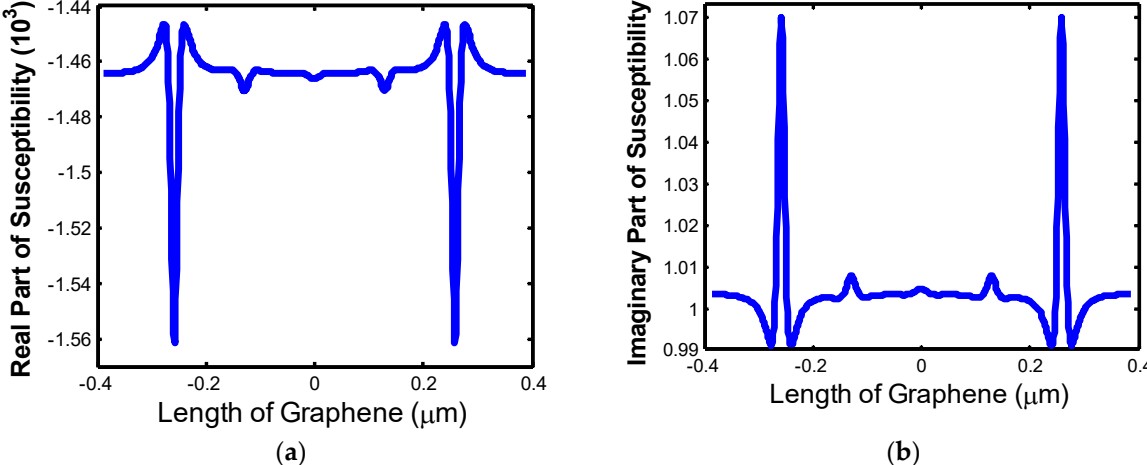

**Figure 9.** The susceptibility for the sagittal case for distribution in whole graphene nanoribbon (**a**) real part, and (**b**) imaginary part, $\mu_c = 0.3$ eV.

By comparing the numerical simulation of the two models (Figures 5–8), we find that the contrast of the grating in the second model is less significant. In other words, the amplitude of the susceptibility is affected using a different model. Therefore, we can adjust the grating structure in the desired pattern by using the superimposed sagittal and Gaussian structures.

As shown in Figure 10, because the changeable chemical potential of the graphene nanoribbon has a significant effect on susceptibility, it can bring significant prosperity to the photonics and Plasmonics industry. The imaginary part represents the losses in the structure, and, with a chemical potential of less than 0.3 eV, the least losses are reported, which is the result of reports and other articles [13,21].

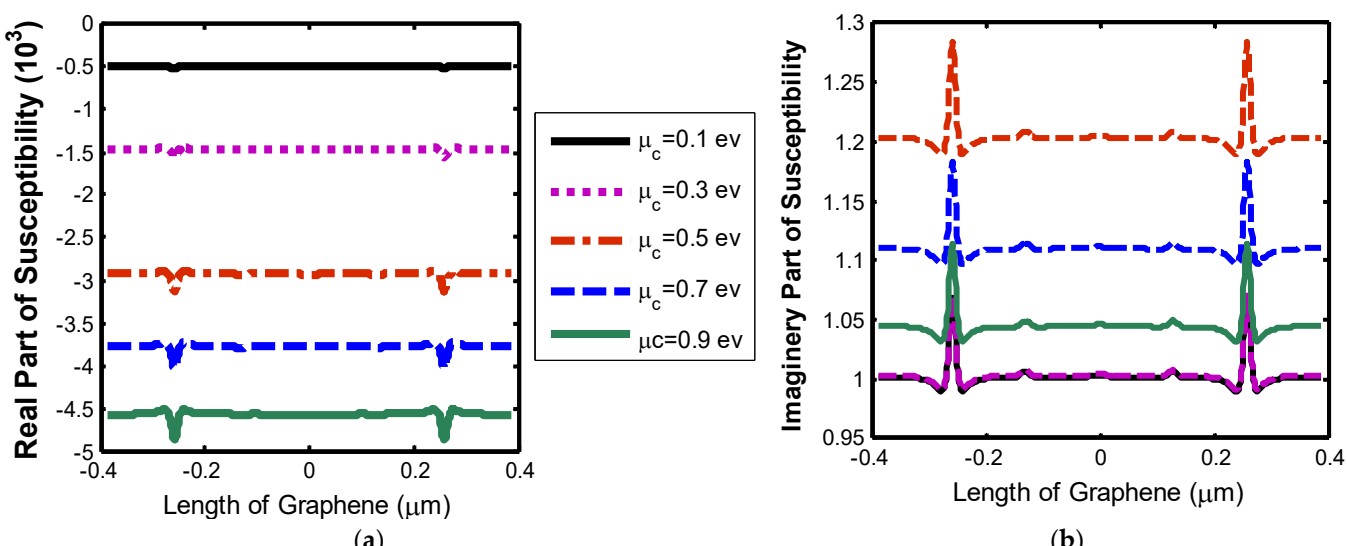

**Figure 10.** The susceptibility for the sagittal case for distribution in whole graphene nanoribbon with changeable chemical potential, (**a**) the real part of the susceptibility and (**b**) the imaginary part of the susceptibility.

## 4. Conclusions

In this work, we used an array of quantum dots on a graphene nanorod to introduce the appropriate lattice structure at the nanoscale. We arranged the quantum dots on the longitudinal axis of the graphene nanoribbons in different shapes and investigated the behavior of the proposed structures based on Columbus's law and the modeling of

surface plasmon polaritons. Theoretically, we obtained the sensitivity of the structure by examining the interaction between quantum dots, and between quantum dots and graphene nanoribbons. Finally, by analyzing the results, we were able to form the desired lattice structures using the dimensions and arrangement of quantum dots.

As expected, these structures have a variety of applications in classical and quantum optical integrated circuits, nanoscale atomic lithography for nanoscale production, coupling coefficient adjustment between waveguides, and the implementation of optical gates, reflectors, detectors, and modulators.

**Author Contributions:** S.A. write the paper and simulated the task. A.R. designed the project conceptually and write and edit the paper and supervise the project. P.M. writes and edits the paper. All authors have read and agreed to the published version of the manuscript.

**Funding:** This research received no external funding.

**Institutional Review Board Statement:** It does not apply to this paper.

**Informed Consent Statement:** It does not apply to this paper.

**Data Availability Statement:** There is no data in this paper to publish.

**Conflicts of Interest:** The authors declare no conflict of interest.

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
