# Peer review of "Interaction between Graphene Nanoribbon and an Array of QDs: Introducing Nano Grating"

_photonics, doi:10.3390/photonics9050348_

Round 1

Reviewer 1 Report

This is a re-submitted manuscript. Compared with its original version, it has been improved. However, I don’t think it is good enough for the publication.

  1. In my previous comments, I suggested the authors to support their conclusion with some experimental results. However, they only give the simulation results in the revised manuscript. How can we check the accuracy of the simulation?
  2. The description of the simulation model is unclear. For instance, I don’t understand how the authors model the QD. Is it regarded as a dielectric nano ball? In equation (3), the polarization of QD is given. Unfortunately, the authors don’t explain the physical details for the parameters. I don’t know what rQD, and E(int)QD- mean. By the way, how do the authors get equation (3)? The electric field must be a vector, but its direction can be seen clearly in equation (3). The authors mentioned they obtained equation from Coulomb’s law. However, the quantum dots have strong quantum effects, I don’t think it is good to model it as a classic particle.

Author Response

Dear Editor

Enclosed is the revised version of the paper submitted for your consideration and publication. We applied all comments in detail and we modified the manuscript. Also, in the following, we present a short response to each question.

Bests

Ali Rostami

Reviewer 2 Report

The manuscript can now be accepted for publication.

Author Response

(The authors gave the same response as above.)

Reviewer 3 Report

The authors answered my previous questions satisfactorily. I recommend the publication in its present form.

Author Response

(The authors gave the same response as above.)

Reviewer 4 Report

There are different types of Graphene nanoribbons, which can be categorized by the presence of band-gap (semi-conducting or metalic) or the chiral angle (armchair, zig-zag, and so on...). Their electronic and optical response of these ribbon can be different. For example, the zigzag ribbon could have magnetic order (Nature 514, 608 (2014)), could experience exotic electron tranport (Nature 506, 349 (2014); PRL 116,236602 (2016)). The authors should comment if the types of Graphene Nanoribbon matters their findings.

Author Response

Dear Editor

Enclosed is the revised version of the paper submitted for your consideration and publication. We applied all comments in detail and we modified the manuscript. Also, in the following, we present a short response to each question.

Bests

Ali Rostami

This manuscript is a resubmission of an earlier submission. The following is a list of the peer review reports and author responses from that submission.

Round 1

Reviewer 1 Report

 In this manuscript, the authors discussed QDs/graphene nanoribbon systems and calculate the absorption coefficient, refractive index and etc. However, I don’t think this manuscript should be published on Photonics due to the following reasons:

  1. Although the authors declared the array of QDs deposited on the graphene nanoribbon, they don’t explain what kind of QDs is used. As shown in Fig.2, the diameter of the QDs is 40 to 50 nm. I doubt whether it can be regarded as QD because the size is too large. On the other hand, the authors have to explain clearly why the QDs are used here. Do you use the quantum effects of QDs for the nano grating? I can’t find any support from the equations listed in the manuscript.
  2. The figures are very unclear and difficult to understand. For example, in Fig.1, I don’t know what the blue ball and the grey sheet represent. Similarly, I don’t understand the meaning of Fig.4 and Fig.6.
  3. As shown in Fig.2, the polarization is quite sensitive to the size of QDs. For the practical fabrication, the QDs size normally has a divergence about 1 nm. The authors have to analyze what will happen in a practical system.
  4. All of the results are obtained from the simulation. They have to be compared with the experimental characterization. Without experimental results, I don’t think these simulation results are very useful.

Reviewer 2 Report

The comments are attached in the word file.

Reviewer 3 Report

In their manuscript “Interaction between Graphene Nanoribbon and an array of QDs: Introducing Nano Grating”, the authors calculated the electromagnetic response of a graphene nanoribbon with an array of quantum dots on its surface. They analysed how the susceptibility of the nanoribbon changes with the size and number of quantum dots, as well as the distance between them. The presented results are novel and will benefit the field of nanophotonics, but the manuscript requires several improvements before the paper can be considered for publication.

[1] I understand that in this study, quantum dots are modelled as spherical dielectric particles characterized by some static dielectric permittivity εd. I have several questions relating to this. Is this permittivity not dependent on the frequency of exciting electromagnetic wave? Also, quantum dots are known to have a noticeable size-quantization of their electronic properties; is it accounted for in this study? Finally, depending on the material, quantum dots can have a different band gap; how does it affect the optical response of the system?

[2] Figure 2 seems rather redundant, as it features three constant functions. It would be sufficient to state these results in the text.

[3] The curve in Figure 3 looks like it has an analytical expression. I suppose that the polarisation of quantum dots might be proportional to their volume (cube of radius), but this needs to be checked. If there is an analytical expression for the polarisation of a single quantum dot, it must be provided in the text.

[4] In Figure 4, it is not clear how quantum dots are arranged on top of the nanoribbon. The figure needs to be improved. It would be more helpful to draw the quantum dots explicitly.

[5] The calculations in Figures 2, 3, and 5 are done using chemical potential of μc = 0.3. First, units must be provided here. Second, the value of chemical potential should also be given in Figures 7–10 (is it the same?). In addition to this, why do the authors use this specific value of chemical potential and not, for example, μc = 0? Finally, the authors need to analyse how the response of the system changes with the chemical potential of a graphene nanoribbon.

[6] Figure 6 is also not very clear with respect to how quantum dots are positioned and what their sizes are. This figure needs improving just like Figure 4.

[7] The caption of Figure 7 seems to be incorrect. The difference between blue and dashed red curves is not explained properly.

[8] In Figures 7–10, it would be helpful to also show the susceptibility of a bare graphene nanoribbon (without quantum dots).

[9] I have problems connecting results in Figure 8 with those in Figures 9 and 10. As the authors show in the last two figures, the susceptibility changes most where the largest quantum dots are. At the same time, Figure 8 shows how the susceptibility changes with the size of the quantum dots. There, it can be seen that the size only affects the broadening of the peaks in the susceptibility plot and not the absolute values of the change. The authors need to elaborate on this issue.

[10] This study is justified by the need of graphene-based plasmonic waveguides for integrated photonic circuits. While the authors present the results of their calculations for a new graphene-based system, I feel that some in-depth discussion, that would connect the results to the overall goal, is lacking. The authors need to add some analysis of how exactly studied graphene nanoribbons with quantum-dot arrays can be used in photonic devices and what makes them promising for such application.

Reviewer 4 Report

I think this article does not add anything of value to the field of photonics. In addition, the article is based just on theoretical models without providing any empirical evidence of the model. Therefore I reject this article for publication.